# Factors predicting parenting stress in the autism spectrum disorder context: A network analysis approach

Khan Buchwald[1], Daniel Shepherd[1]*, Richard J. Siegert[1], Matthieu Vignes[2], Jason Landon[1]

**1** Faculty of Health and Environmental Sciences, Auckland University of Technology, Auckland, New Zealand, **2** School of Mathematical and Computational Sciences, Massey University, Auckland, New Zealand

* daniel.shepherd@aut.ac.nz

## Abstract

Elevated levels of parenting stress have been reported in parents raising an Autistic child. Previous studies have identified a multitude of predictors of parenting stress, including both child-related and parent-related factors, though findings across studies are not always in agreement. In the present study we investigate the determinants of parenting stress using a Network Analysis approach, which is then used to inform a subsequent structural equation model. New Zealand parents (*n* = 490) of a child diagnosed with Autism Spectrum Disorder (ASD) provided data on their Autistic child (e.g., ASD core symptoms, problem behaviours) and themselves (i.e., parenting stress). The analysis revealed that both child and parent demographic factors were poor predictors of parenting stress, while the child's current language and communication ability were correlated with diagnostic age and parenting stress. An earlier diagnostic age, in turn, suggested better behavioural and emotional outcomes for children. Overall, the Network Analysis showed itself to be an informative approach to understanding parenting stress in the ASD context. Findings further advocate for the implementation of ASD-related and language-related interventions as early as possible, and that language delays during early infancy justify prompt clinical assessment.

## Introduction

The fifth edition of the Diagnostic and Statistical Manual 5-TR [1] describes Autism Spectrum Disorder (ASD) as a neurodevelopmental disorder characterised by (i) persistent deficits in social behaviour and communication across multiple contexts, and (ii) restricted, repetitive patterns of behaviour, interests or activities. These two diagnostic criteria must be evident in childhood or adolescence, result in substantial impairments in functioning in daily life, and not be better explained by any alternative diagnosis. With few evidence-based treatment options currently available, parents are usually at the forefront of ASD symptom management, a role that typically extends far beyond the normal parenting period. Studies have consistently shown that parenting an Autistic child is associated with even greater challenges than those

**Data availability statement:** All relevant data are within the manuscript and its Supporting Information files.

**Funding:** The author(s) received no specific funding for this work.

**Competing interests:** The authors have declared that no competing interests exist.

of parents of children with other developmental disabilities [2], and is a strong predictor of poor psychological wellbeing [3,4]. Systematic reviews of quality of life (QoL) in parents of an Autistic child report reduced QoL compared with parents of neurotypical children and population norms [5].

Shepherd et al. [6] reported that, in their sample of 658 New Zealand parents of an Autistic child, most participants exhibited clinical levels of psychiatric distress. Across the ASD literature, parenting stress has consistently been identified as the dominant predictor of compromised parental wellbeing [7–10]. Parenting is inherently challenging, even when dealing with normative family events typically experienced by all parents [11], and even more so when a child has additional needs. When the demands associated with the role of parenting surpass the available resources at the parent's disposal then parenting stress results [12], which embodies feelings of distress or discomfort and induces negative emotions. Furthermore, parenting stress has a direct effect on family QoL (FQoL) and is associated with undesirable transactional effects [13], though emerging evidence suggests that parenting stress levels may mediate the relationship between child core autism symptoms and FQoL [14].

The cause of parenting stress is multifactorial [15,16]. Efforts to elucidate the predictors of parenting stress have focused primarily on both child-related and parent-related factors [17], and external factors such as social supports [18]. The contemporary literature indicates that parenting stress is associated with both child gender and age [10], the severity of core symptoms [19,20], and the presence of comorbid problem behaviours [21,22]. Age and gender [23,24], educational attainment, and family size are parental predictors of parenting stress, along with experiences of social stigma [25,26]. Conversely, both informal and formal social supports and social media have been shown to attenuate parenting stress [15,27,28]. In relation to parent and child age, the 'wear-and-tear' hypothesis advances the notion that parenting stress will increase over time due to fatigue and a reduction in formal and informal support. In contrast, the coping hypothesis states that with time parents develop resilience and learn to optimise sources of support. However, definitive support for either hypothesis has thus far been elusive [29,30].

The severity of ASD core symptoms is a reliable predictor of impact on parents, with more severe symptoms associated with higher stress levels [20,31,32]. However, Benson [33] reports that the severity of comorbid problem behaviours and the absence of prosocial behaviours are better predictors of parenting-stress than overall ASD symptom severity. While some studies have reported no statistical relationships between core ASD symptoms and parenting stress beyond social deficits [35], others describe significant positive relationships between parenting stress and both severity of ASD symptoms and problem behaviours [35,36]. Clearly, further clarification of the relationship between ASD core symptoms, comorbid problem behaviours, and parenting stress is required.

Managing parenting stress in the ASD context is important for both parents and their children. Caring for an Autistic child is associated with additional physical and emotional demands, which in turn can place strains on parents and their interpersonal relationships. High levels of parenting stress can also adversely affect the Autistic child [13,37], counteracting therapeutic gains [7], and increasing the risk of child abuse or neglect [38]. As such, it is important for both the parent and the child to have strategies in place to minimise parenting stress, and this necessitates identifying the risk and protective factors of parenting stress. Previous efforts to document the predictors of parenting stress in the ASD context have typically employed correlation or regression analysis, with the latter frequently focusing on variables that moderate or mediate the relationship between ASD-related factors and parenting stress. In the present study we employ a Network Analysis (NA) approach, with Directed Acyclic Graphs (DAGs) allowing us to explore which variables are most likely to explain high levels of parental stress in parents of an Autistic child.

The goal of NA is to portray the relationships among the symptoms (variables; nodes) of a disorder or disorders by inferring their connections (edges) as a network. Nodes could refer to items in a survey, unobserved latent variables, or observed variables. Edges are based on a statistic, such as a partial correlation, and represent conditional relationships among the variables [39]. The application of NA to understanding psychological disorders is a recent development but its popularity is growing rapidly [40], and the use of NA is emerging in the ASD literature [41]. For example, Li et al. [42] used NA to retrospectively identify the most common comorbid conditions based on ICD-10 codes in the case records of 1488 people diagnosed with ASD. Pino et al. [43] used NA to explore the associations among domains of social cognition in children with ASD and compared these to the social cognition domains network of typically developing children.

The relationships among ASD symptoms has also been examined using networks of symptom scales. Wang et al. [44] used NA to explore the relationships of core social symptoms of ASD and associated comorbidities in a sample of 474 children with ASD. They reported that social communication deficits were the most central symptoms in the network and observed a direct relationship between anxiety problems and core ASD symptoms such as restricted interests and repetitive behaviours. Wang et al. concluded that the widespread positive correlations between core and 'non-core' (i.e., comorbid) symptoms of ASD suggest that intervening to target associated symptoms (e.g., internalising or externalising behaviours) might potentially improve core ASD symptoms. Also using NA, Montazeri et al. [45] nominated communication and social abilities as potential treatment targets during early development, noting a strong interdependence between autism-related verbal and non-verbal communication deficits.

## The present study

Child ASD-related symptoms have been found to be poor predictors of parental psychological wellbeing [46], and instead parenting stress has emerged as a key predictor. Better parenting results in better child outcomes, and identifying factors related to parenting stress allow for more effective and targeted parent-based interventions. By acknowledging reciprocal relationships between variables, and that a variable may have proximal or distal effects on other variables, NA can better represent the structures underlying complex systems. The particular network approach employed in the current study involved a Bayesian network analysis comprising DAGs. The use of DAGs has become widespread in psychology and epidemiology to discern dependency or causal relationships among variables in observational datasets. In addition, we combined this method with structural equation modelling (SEM) to quantify the goodness of fit of the models obtained from our NA, and to estimate the strength of the relationships between elements of the network.

## Method

### Participants

The sample (N = 490) consisted of New Zealand parents of an Autistic child, with data collected between June 2022 and September 2022. The mean age of participants was 45.3 years (SD = 9.5), the majority were female (n = 432), most identified as European (n = 383) compared to Māori (n = 47), and the majority had a university degree (n = 259). Participants reported that their Autistic child was more likely to be male (n = 371), and that over half were diagnosed by a paediatrician (n = 282). Of note, approximately half of the children were diagnosed with ASD at ages less than five years, and approximately half of children were diagnosed with ASD at ages greater than or equal to five years. Table 1 provides a comprehensive

**Table 1.  Parent and child demographic profile.**

| Demographic | *n* | % | Missing |
|---|---|---|---|
| Sex (Parent) | | | 2 |
| Male | 56 | 11.5 | |
| Female | 432 | 88.5 | |
| Sex (Child) | | | 6 |
| Male | 371 | 76.7 | |
| Female | 113 | 23.3 | |
| Age (Parent) | | | 2 |
| <45 | 240 | 49.2 | |
| 45 + | 248 | 50.8 | |
| Age (Child) | | | 6 |
| <13 | 266 | 55 | |
| 13 + | 218 | 45 | |
| Ethnicity (Parent) | | | 1 |
| European | 393 | 80.4 | |
| Māori | 47 | 9.6 | |
| Other | 49 | 10 | |
| Education (Parent) | | | 2 |
| No University Degree | 229 | 46.9 | |
| University Degree | 259 | 53.1 | |
| Marital Status (Parent) | | | 12 |
| Single | 104 | 78.2 | |
| Relationship | 374 | 21.8 | |
| Number of Children (Parent) | | | 10 |
| 1 | 168 | 35 | |
| 2 + | 312 | 65 | |
| Age of ASD Behaviour | | | 1 |
| <2 | 239 | 48.9 | |
| 2 + | 250 | 51.1 | |
| Age at diagnosis | | | 16 |
| <5 | 260 | 54.9 | |
| 5 + | 214 | 45.1 | |
| Diagnosing Clinician | | | 0 |
| Paediatrician | 282 | 57.6 | |
| Psychologist | 131 | 26.7 | |
| Other | 77 | 15.7 | |

Note: *n* = number of participants; % = percent of sample; Missing = number of missing values for this variable.

participant and child profile for the variables included in the Network Analysis. Ethical approval for this study was obtained (Auckland University of Technology Ethics Committee: 21/218).

## Measures

### Parenting stress

Parenting stress was estimated using the 18-item Parenting Stress Scale (PSS) [47]. The PSS covers positive themes of parenthood such as personal growth and emotional benefits, and

negative consequences such as restricted lifestyle and strains on resources. A response to each item is solicited using a five-point Likert scale ranging from 1 (strongly disagree) to 5 (strongly agree). For the current study the PSS was reduced to four subscales: Parental stressors, parental satisfaction, lack of control, and lack of rewards [47, 48].

### Child autism symptoms

The Impact dimension of the Autism Impact Measure (AIM) [49] was used to gauge parents' perceptions of their child's ASD symptom severity. The 25 parent-rated questions of the impact dimension assess how autistic traits affect the daily functioning of the child. The 25 items are summed to produce four subscales: Restricted/Ritualized Behaviours (8 items), Odd/Atypical Behaviours (5 items), Communication/Language Impairment (5 items), and the absence of Social-Emotional Reciprocity Skills (7 items). Each item was considered using a five-point Likert scale that ranged from 1 (not at all) to 5 (severely), and item descriptors can be found in Kanne et al.'s [49] Table 4 (p. 175).

### Child problem behaviours

The Strengths and Difficulties Questionnaire (SDQ) [50] was employed to assess internalising and externalising behaviours displayed by the participant's Autistic children. The SDQ has 25 items distributed across five subscales: Hyperactivity (e.g., restlessness, easily distracted), Emotional Symptoms (e.g., worry, tearfulness), Conduct Problems (e.g., disobedience, dishonesty), Prosocial Behaviours (e.g., empathy, sharing), and Peer Problems. Participants responded to items probing their child's everyday behaviour using one of three response options: "not true", "somewhat true", or "certainly true". Due to a programming error one of the items in the emotional symptoms subscale (*Often unhappy, down-hearted or tearful*) was not analysed, and the subscale was computed without this item.

## Procedure

Recruitment involved advertisements placed in two New Zealand autism support group newsletters. Adverts contained information on the study's purpose, an invitation to participate if inclusion criterion were satisfied, and a link to the survey itself. The survey was presented on Qualtrics, and prior to survey items being presented the participants were informed that participation was voluntary, that the privacy and anonymity of participants was assured, and that they could stop responding at any time. Additionally, they were informed "*By completing the questionnaire, you are expressing your consent to participate in this study.*"

## Analysis

All data analyses were conducted in R version 4.2.2. The Bayesian networks were performed using the package bnlearn version 4.7.1 [51] and structural equation modelling through the package lavaan version 0.6-16 [52].

### Preliminary analyses

Descriptive analyses included means and standard deviations for scalar items, while Cronbach's alphas ($\alpha_c$) were computed as part of the reliability analysis. When appropriate, missing data (<5%) was imputed using the random forests algorithm using the R package missForest version 1.5 [53].

Education, marital status, number of children, age of child when ASD behaviours first occurred, and age at diagnosis were transformed into dummy variables by estimating an even

split for these variables. An 'other' category was used for ethnicity if a person did not identify as European or Māori, or if the person identified as having multiple ethnicities without specifying which ethnicity they identified with. Furthermore, we aggregated all diagnosing clinicians that were not paediatricians or psychologists and labelled them as 'other'. If participants reported multiple clinicians or a service, this was labelled as 'other'. All demographic variables were transformed to dummy variables to implement the structural equation model (SEM) in lavaan [52]. Dummy variables were ordered so the most frequent category was listed as 0 and the least frequent category was labelled as 1. Although these variables were included in the models presented in the supplementary materials, only the demographics: age at diagnosis, age of parent, age of child, and the ethnicity variable 'European or other' were included in the network reported below.

## Network analyses

All network analyses were implemented using the bnlearn package [51]. A constraint-based and score-based algorithm was implemented on the data to reconstruct the Bayesian network, with the algorithm having the best Bayesian Information Criteria (BIC) score subsequently selected. Hill climbing and Tabu algorithms in the bnlearn package [51] converged to the same network, with both score algorithms having the best BIC. In addition, a network with and without a blacklist was implemented, where a blacklist is a restriction of the edge space in the reconstructed network [51]. An edge, between parent node and child node, that has been blacklisted will not appear in the reconstructed network. Firstly, all demographics were restricted to be parent nodes of assessment variables only, as in the package bnlearn [51], categorical variables such as demographics are restricted to be parent nodes of continuous variables and therefore are blacklisted by default. Secondly, in the 'Blacklist' networks, the subscales of the parental stress scale were restricted so that they could not be parent nodes of subscales of the SDQ and AIM scales, ensuring that parenting stress subscales were the outcome nodes in the network. This approach aligns with the hypothesis that parenting stress is an outcome of the other variables in the network. The networks without the blacklist do not have such restrictions, and are placed in the supplementary materials section.

Because both discrete and continuous variables were included in the network a hybrid Bayesian network was implemented, assuming a conditional Gaussian distribution rather than a multinomial or Gaussian distribution. The hybrid Bayesian learning algorithm *tabu* from the bnlearn R package [51] was used to optimise the network using BIC as the model fit index. Some variables were not connected in the original network (i.e., had no edges) and these were removed sequentially from the model until the network formed a unique connected component. Once the Bayesian networks were implemented, it emerged that many demographic variables were not dependent or conditionally dependent on nodes representing the assessment subscales. Subsequently, a model excluding these demographics was implemented, based on the model fitting criteria Confirmatory Fit Index (CFI) and Root Mean Square Error Adjusted (RMSEA), and these other models can also be viewed in the supplementary materials section.

The centrality statistics *betweenness* (how well a node acts as a connecting node based on the number of edges through that node to other nodes), *closeness* (how close a node is to other nodes using the average partial correlation in the paths to or from that node), and *degree* (the number of connections to or from that node) [54] were calculated and standardised (*re*: supplementary materials). In addition to centrality statistics, an averaged network for the model was created with a blacklist and selected demographic variables removed [55,56], also presented in supplementary materials. To calculate an averaged network, 10,000 bootstrapped

samples are generated, and a network is fitted onto each of these sample. The networks are then averaged together. Averaged networks were generated as they tend to have more predictive validity than networks produced on the complete data [57].

## Structural equation modelling (SEM)

To further examine the network properties a SEM using the lavaan package was produced [52], using the selected network, and the network with a blacklist imposed and with some demographics removed. From these SEMs, chi-square tests of model fit, regression coefficients, the *p*-values of the edges between observed variables in the SEM, and fit indices CFI and RMSEA, were extracted. CFI values above.95 and RMSEA below approximately.6 were used to indicate a good model fit [58]. No latent variables in this study were added to the model as the purpose of the structural modelling was to test whether the reconstructed network was preferred over a null model.

## Network model querying

Finally, the hybrid Bayesian network was queried to explore the nature of the association between variables. While these queries are not formal hypotheses as used in inferential testing, they do provide probabilities that can be used to make clinical inferences. These probabilities are conditional, for example, given that the Age at Diagnosis is less than five years of age, what is the probability that scores on Conduct Problems is greater than 3? These queries are based on Monte Carlo simulations, and the *hence* function in the package bnlearn uses approximate inference to obtain these probabilities [51].

# Results

## Preliminary analyses

The summary statistics for the scalar measures included in the network are presented in Table 2.

## Network analyses

Fig 1 shows the Bayesian network structure for the blacklisted network, where the variables are represented as nodes, and the lines between them as edges. This network suggests that Age at Diagnosis was more often the parent node to the subscales of the AIM, SDQ and PSS compared to other demographic variables. In NA, a parent node refers to a node that has a directed edge pointing to another node, known as its child node. Age at Diagnosis did not have any parent nodes and had four edges directed away from it. The Lack of Control subscale from the PSS is the outcome variable of the remainder of the relationships of the network. Interestingly, a path was identified through Parental Stress to Parental Satisfaction to Lack of Control.

## Structural equation modelling

Fig 2 is the path analysis of the reconstructed hybrid Bayesian network, documenting a number of significant associations between the variables within the network. In particular, Age at Diagnosis is a significant predictor of the Communication/Language subscale, the Conduct Problems subscale, the Age of the Parent, whether the parent child's ethnicity was European or other, and the Emotional Problems subscale of the SDQ. In addition to this, the Parental Rewards subscale was a predictor of lack of control, after controlling for the Parental Stressors and Parental Satisfaction subscales. Parental Satisfaction, where higher scores indicate

**Table 2. Summary statistics of the AIMS, SDQ, and PSS.**

| Factor | M | SD | Min | Max | Missing |
|---|---|---|---|---|---|
| AIM | | | | | |
| Restricted/Ritualised Behaviour | 27.3 | 6.0 | 8 | 40 | 13 |
| Communication/Language | 14.6 | 5.5 | 5 | 25 | 8 |
| Social-Emotional Reciprocity | 18.2 | 5.5 | 7 | 35 | 9 |
| Odd/Atypical Behaviour | 15.7 | 4.5 | 5 | 25 | 10 |
| SDQ | | | | | |
| Emotional Problems* | 4.1 | 2.3 | 0 | 8 | 21 |
| Conduct Problems | 2.9 | 2.1 | 0 | 10 | 17 |
| Hyperactivity | 6.5 | 2.7 | 0 | 10 | 19 |
| Peer Problems | 5.8 | 2.1 | 0 | 10 | 18 |
| Prosocial | 3.8 | 2.5 | 0 | 10 | 20 |
| PSS | | | | | |
| Parental Rewards | 13.4 | 4.8 | 6 | 30 | 33 |
| Parental Stressors | 21.4 | 5.5 | 6 | 30 | 29 |
| Lack of Control | 8.3 | 3.3 | 3 | 15 | 29 |
| Parental Satisfaction | 7.8 | 2.7 | 3 | 15 | 28 |

Note: AIM = Autism Impact Measure; SDQ = Strengths and Difficulties Questionnaire; PSS = Parental Stress Scale; M = Mean; * = Item 3 not available for this sub-scale.

greater dissatisfaction, is positive, so as Parental Satisfaction increases so does Lack of Control. Furthermore, there is a significant increase in reported Conduct Problems, and a significant decrease in the Communication and Language Impairments subscale, as the Age at Diagnosis increases from younger than 5 years to 5 years or older.

Fit statistics for the model presented in Fig 2 are reported in Table 3. We conducted hypothesis tests to identify if the model implied variance covariance matrix is significantly different to the observed variance covariance matrix. As all four models are significantly different, this indicates that the network model does not represent the true underlying model for the data. However, the SEM models with some demographics removed had a much better fit in terms of CFI and RMSEA, while the models with a blacklist had poorer fits than the models without a blacklist (*re*: CFI, RMSEA, BIC, and AIC). This is expected as the model without the blacklist has less constraints to find a local optimum with a better fit. Note that lower log likelihood, BIC and AIC values indicate a better fit.

## Network model querying

Table 4 presents the results of selected queries made to the Bayesian network with a black-list (*re*: Fig 1). The table shows the probability of a second variable (Variable 2) occurring given the condition made on another (Variable 1). Here, larger differences in the values in the probability column for the same variables indicates that Variable 2 is more conditionally dependant on Variable 1, compared to when the probability does not change between the two categories. For example, if a child is aged older than 5 years there is a.554 probability that scores on the Conduct Problems subscale will be greater than 3. However, if a child is diagnosed at an age less than 5 years the probability of scores on the Conduct Problems subscales being greater than 3 is.383. The probabilities of scores on the PSS subscales were considerably different depending on the scores on the AIM Communication and Language subscale. For Communication and Language scores below 14, there was a lower probability

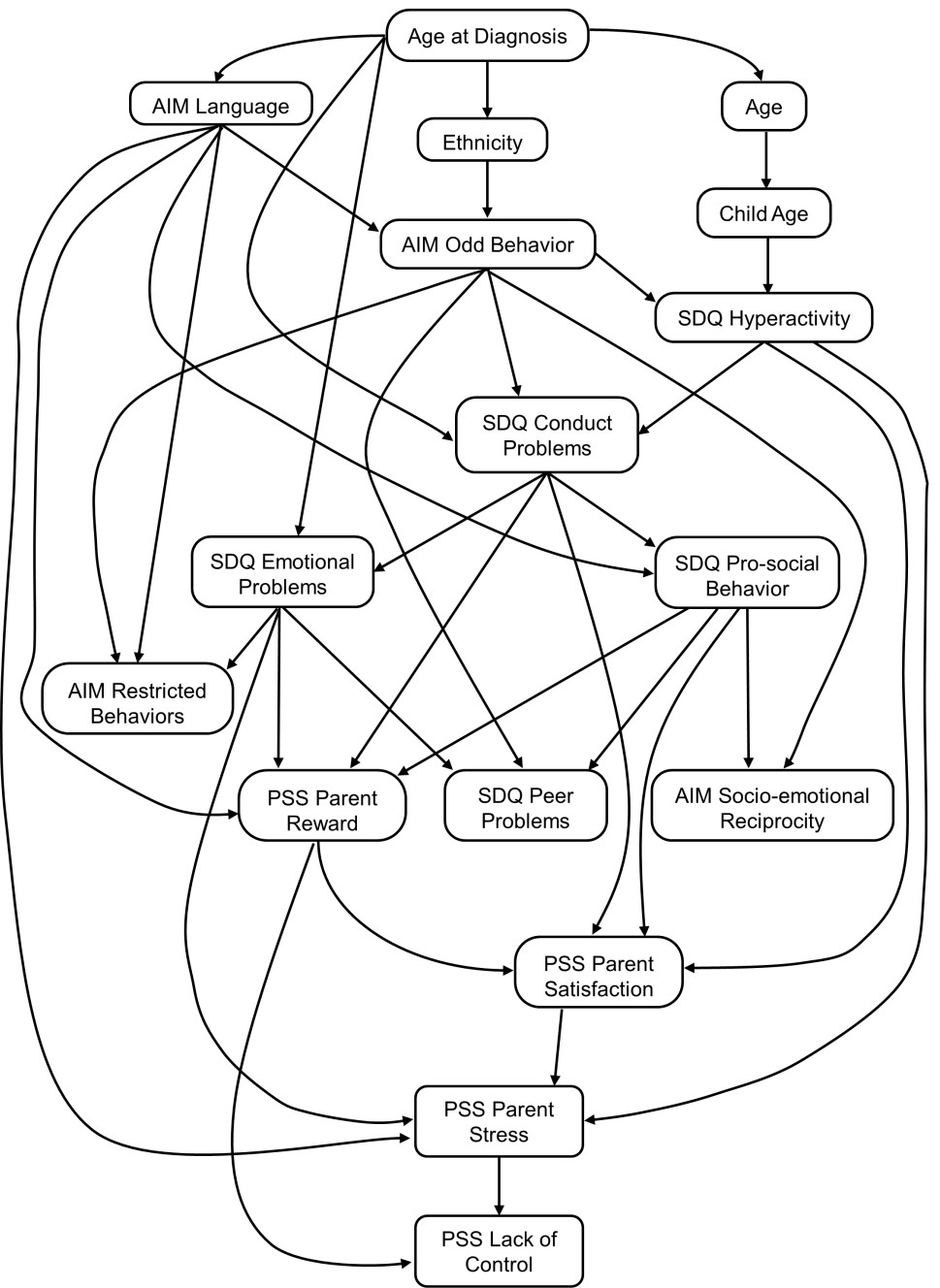

**Fig 1.** Hybrid Bayesian network without blacklist and excluding some demographics.

of Parental Rewards scores (.446) compared to Communication and Language scores 14 and above (.546). Further, there was a higher probability of Parental Stressors scores greater than 21 if the child scored above 14 on the AIM Communication and Language subscale, compared to scoring lower than 14 on Communication and Language subscale. This is also true for conditional probabilities between Communication and Language, and Lack of Control and Parental Satisfaction.

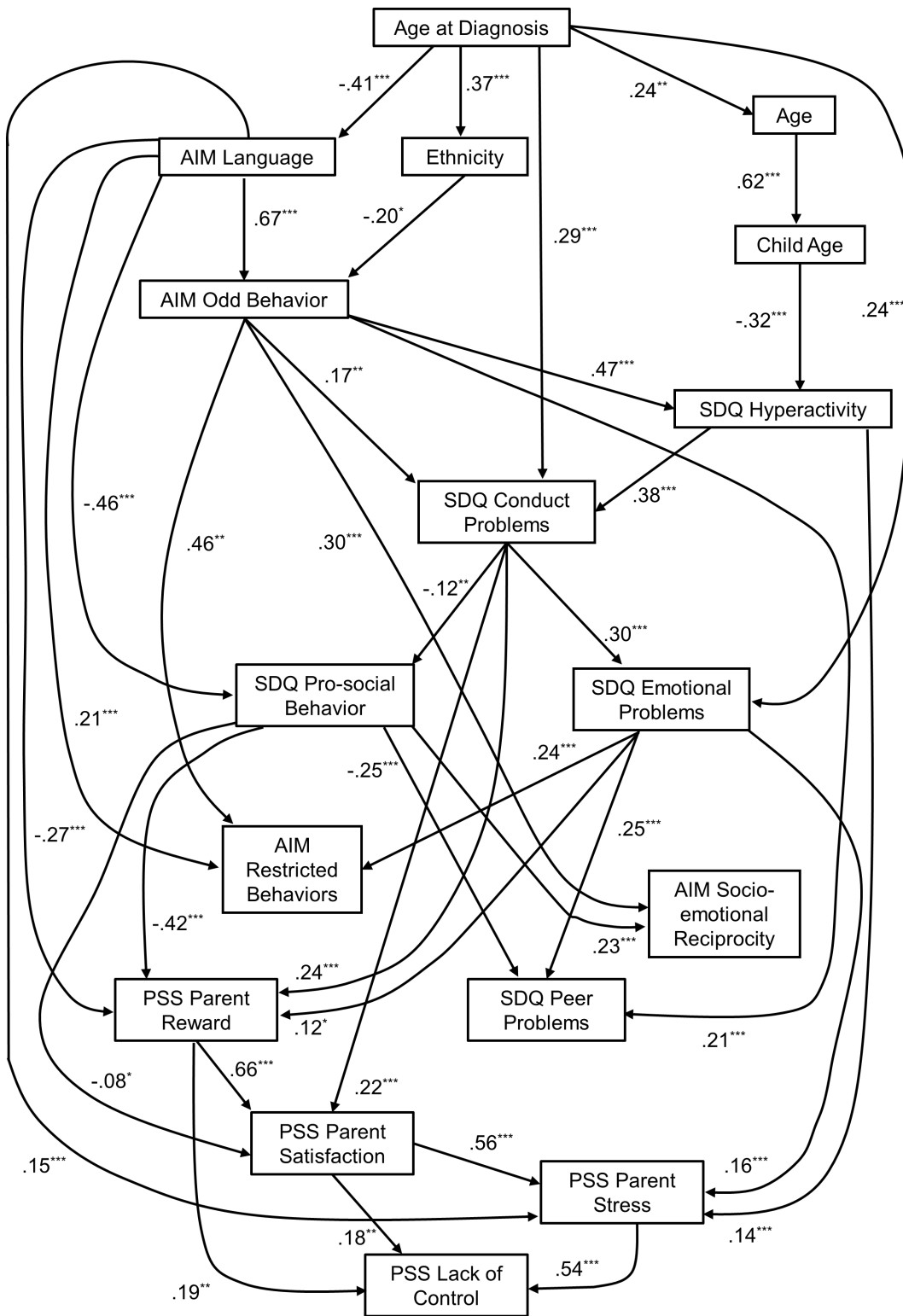

**Fig 2.** Path analysis of hybrid Bayesian network presenting standardising path coefficients.

**Table 3. Structural equation model fits.**

| Model | SEM | | | | | | Bayesian Network | | |
|---|---|---|---|---|---|---|---|---|---|
| | CFI | RMSEA | Chi Sq | df | p | Adj p | Log-Likelihood | BIC | AIC |
| No Blacklist, includes demographics | .508 | .215 | 4883 | 206 | <.001 | <.001 | -18864 | -19149 | -18956 |
| Blacklist, includes demographics | .510 | .217 | 4859 | 203 | <.001 | <.001 | -18863 | -19157 | -18958 |
| No Blacklist, excludes demographics | .983 | .034 | 155 | 99 | <.001 | .005 | -16666 | -16899 | -16741 |
| Blacklist, excludes demographics | .977 | .041 | 166 | 92 | <.001 | <.001 | -16662 | -16904 | -16740 |

## Discussion

This study further extends the parenting-stress literature by employing a Network Analysis approach. The inductive/deductive approach used afforded a number of testable hypotheses that replicate previous findings in the literature [e.g., 16], in addition to contributing some novel findings. These findings will now be summarised.

### Not all variables are useful predictors of parenting stress

It was noted that the network model better accounted for the data when the following demographic variables were excluded: ethnicity (Māori compared to European), parental education, marital status, child gender, diagnosing clinician, and the number of children in the family. A lack of statistical significance for ethnicity, child gender, and parent education has been reported previously in a comparable population [6], as has a lack of covariance between sibling number and parenting stress [15]. Likewise, the lack of a link between parenting stress and parent reports of their child's age when ASD-like behaviours first emerged has precedence [59]. Thus, as these variables were not useful predictors of the subscale scores it can be argued that evidence is accumulating across the literature that a number of parent- and child-related characteristics are neither direct attenuators nor agitators of parenting stress.

**Table 4. Queries to the Hybrid Bayesian Network (re: Fig 1).**

| Variable 1 | Value 1 | Variable 2 | Value 2 | Probability |
|---|---|---|---|---|
| Age at Diagnosis | <5 | Conduct Problems | >3 | .383 |
| Age at Diagnosis | 5+ | Conduct Problems | >3 | .554 |
| Age at Diagnosis | <5 | Communication/Language | <14 | .299 |
| Age at Diagnosis | 5+ | Communication/Language | <14 | .665 |
| Age at Diagnosis | <5 | Parental Stressors | >21 | .523 |
| Age at Diagnosis | 5+ | Parental Stressors | >21 | .533 |
| Age at Diagnosis | <5 | Prosocial | >3 | .412 |
| Age at Diagnosis | 5+ | Prosocial | >3 | .306 |
| Communication/Language | <14 | Parental Rewards | <13 | .446 |
| Communication/Language | 14+ | Parental Rewards | <13 | .546 |
| Communication/Language | <14 | Parental Stressors | >21 | .418 |
| Communication/Language | 14+ | Parental Stressors | >21 | .598 |
| Communication/Language | <14 | Lack of Control | >8 | .444 |
| Communication/Language | 14+ | Lack of Control | >8 | .553 |
| Communication/Language | <14 | Parental Satisfaction | <8 | .468 |
| Communication/Language | 14+ | Parental Satisfaction | <8 | .515 |

### Seeking a diagnosis may be getting easier

The analysis indicated that if the age of the parent is greater than 45, compared to younger than 45, more children were diagnosed with autism at age 5 years or older. This finding can potentially be explained by a number of factors. Firstly, it could be argued that a generational effect exists in which older parents initially avoided seeking a diagnosis due to factors such as social stigma or a lack of awareness. Secondly, older parents in our sample may have higher functioning children than those below the age of 45. Lastly, and more credible given both societal and health delivery changes in New Zealand, barriers to obtaining a diagnosis may have lessened due to the growing evidence that early intervention equates to better later life outcomes for Autistic children.

### Communication and language impairment likely determines age at diagnosis

Several studies [59–62] have reported that language impairment is the dominant predictor of child age at the time of the ASD diagnosis. Specifically, as children age and fail to reach developmental milestones they are more likely to be professionally assessed, and the current results concur with those reported by Meads et al. [59] who utilised the same AIM scale as the current study. As such, any infant exhibiting a language delay may benefit from an appropriate screen to either confirm or exclude a diagnosis of ASD.

### Communication and language impairments are the dominant predictors of parenting stress

Communication and Language Impairment AIM scores less than 14 reduce the probability of high Parental Stressors and Lack of Control scores, whereas scores between 15 and 25 are associated with increased probability of lower Parental Rewards and Satisfaction. This finding is expected, as increased communication between parent and child can attenuate child problem behaviors and increase intervention effectiveness. Of relevance, Shepherd et al. [24] reported a positive relationship between child language ability and both parental free time and family support, and of those parents abandoning speech language therapies, only 12% did so because they felt the intervention was ineffective.

### Communication and language impairment is an important moderator

Further to Fig 2, language ability can be seen to moderate the relationship between Age at Diagnosis and the following: a. Prosocial Behaviour, b. Odd Behaviour, c. the Parental Stressors subscale of the PSS, and d. the Parental Reward subscale of the PSS. The age at which an ASD diagnosis is conferred is important in the neurodevelopmental context, as earlier assessment and diagnosis affords earlier intervention. As such, a direct relationship between Age at Diagnosis and later ASD-related traits would be expected [63], and on account of the positive relationship between ASD-traits and parenting stress [16,64], a significant relationship between parenting stress and Age at Diagnosis has been hypothesised but has not always been supported [59]. In terms of moderation, children with early diagnoses exhibit improved language and communication function, which in turn is related to improved prosocial behaviour and reduced odd behaviours, and in turn less parenting stress and greater parenting rewards. Consequently, it is expected and reported that ASD-related interventions typically focus on improving language and social communication skills [65].

### Age at diagnosis predicts future conduct and emotional difficulties

Children diagnosed below the age of five years have a reduced probability of severe conduct and emotional problems later in life. This finding indicates that early intervention may not

only be effective in reducing core ASD symptoms above and beyond those reductions related to aging, but may also be effective in reducing some problem behaviors. Of relevance to emotional problems, the label "ASD – without intellectual or language impairment" has increasingly been applied to cover the deletion of the Aspergers diagnosis, which was characterised by milder ASD-related symptoms and an absence of language delays. Thus, for children that exhibit a pattern of attenuated ASD symptoms, delays in diagnosis will be more likely as language milestones are satisfied and experiential factors such as schooling and social activities progress as normal. However, reduced inter-personal skills can make such children the targets of bullying and victimisation at school, and hence more vulnerable to severe depression and anxiety, and thoughts of suicidal ideation [66].

## Control may be a key factor for parents caring for a child with ASD

The Parental Satisfaction, Parental Stressors, and Parental Rewards subscales were predictors of the Lack of Control subscale, which emerged as the outcome variable of the network. In the current context control can be linked to the concept of self-efficacy, a parent's self-belief in their ability to effectively manage and support their child with a disability [14,67]. A parent that perceives a lack of control as a caregiver may withdraw from taking an active role in their child's upbringing, avoiding interaction, distancing themselves from parental responsibilities such as engaging required interventions for the child, and expending little effort to minimise problem child behaviours [68]. The consequence of a lack of control, or 'agency', may result in both decreasing parenting confidence and competence, potentially reversing any positive gains from treatment, and negatively impacting the mental health of both parent and child. As such, targeting parenting self-efficacy in parent-based therapy has been identified as a priority by some [14,69].

## Network parameters are informative

In terms of centrality statistics, Conduct Problems and Hyperactivity are the largest modifier variable, as edges pass through parental stressors to other nodes. Parental stressors possess the highest relationship with adjacent nodes, hence interventions targeting distress among adults are likely to have the highest impact on these adjacent nodes, compared to other relationships in the network. Odd Behaviour and Conduct Problems have the highest number of edges, and so changing these variables will likely impact the network structure. Communication and Language had the highest number of outgoing edges indicating it is a predictor of many variables, including the PSS subscales. Interestingly, two previous studies have indicated that language and communication impairment is not significant contributor to parenting stress [34,70], and instead problem behaviours (i.e., externalising or internalising behaviours) are better predictors. However, examination of our Fig 2 suggests that language impairments and problem behaviours may both be predicting parenting stress, that there may be potential interactions between the two, and that a combination of core symptoms and problem behaviours may ultimately determine levels of parenting stress, rather than any single child-related characteristic.

## Strengths and limitations

A strength of the current study is its analytical approach. Network Analysis is a relatively new approach in ASD research, and Directed Acyclic Graphs (DAGs) even more so. However, DAGs are becoming more widely accepted due to their ability to present complex inter-relationships among multiple observable and latent variables in a clear and intuitively meaningful form, while controlling for confounding variables in modelling complex dependency relationships in observational studies [71]. As a cross sectional study, the methodology of our study is not sufficient to infer causal associations between variables in the network,

and instead, examines the dependency relationships between variables [72]. Pertinently, it cannot be discounted that a third variable could be causing the relationships found in the network. However, the current study expands the NA approach further by then testing the model using SEM to determine how well the hypothesized model fits the data and using path coefficients to quantify the strengths of the edges or connections between all pairs of variables.

There are several caveats that need to be considered when interpreting the findings of the current study. First, child ASD-related characteristics were obtained using parent, as opposed to clinician, ratings. Whether this approach should be perceived as a methodological strength or weakness is uncertain, as arguably parents are better placed to judge their child's functional limitations as they spend more time with the child and are not constrained to observe in clinical settings [73,74]. Second, the majority of participants were well-educated females, of European decent, and currently in a relationship. This demographic bias in the sample, characterized by a predominance of highly educated European mothers, limits the generalizability of the study's findings, and this overrepresentation may not reflect the experiences or outcomes of individuals from more diverse racial, ethnic, educational, or socioeconomic backgrounds [13]. In terms of parenting an Autistic child, studies have repeatedly shown that mothers typically experience greater care-related stress than fathers [9,10]. Third, the cross-sectional nature of the study does not afford definitive conclusions of cause and effect, even though many of the bivariate relationships can be interpreted as being intuitive. Fourth, though the sample size is relatively large for a study of this type, recruitment through national autism associations means that the representativeness of the sample cannot be assessed. Further, the exclusion of a single item from the SDQ emotional problems subscale may have adversely affected the psychometric properties of that subscale. Finally, some studies have highlighted potential short-comings with the implementation of the parenting stress index [47] in ASD-related studies [75], and future studies could be designed to directly compare the various parenting stress measures on offer.

## Conclusion

The current study sought to elucidate the risk factors associated with stress when parenting a child with ASD. Arguably the most striking finding of the current study was the degree to which the age of the child at diagnosis was conditionally linked to the severity of several child behaviours and negative parental outcomes. Impairments in communication and language may make ASD more readily identifiable in children, and therefore more likely to result in a timely diagnosis. Thus, one approach to avoid diagnostic delay could form around better information to parents of the risk of language delay and the need to seek assessment. The current findings can also facilitate clinicians in supporting those parenting an Autistic child, specifically in reinforcing parent's perception of self-efficacy and control. Future studies intending to extend the current model in the parenting stress arena should consider including formal and informal support mechanisms, alongside parent coping strategies.

## Supporting information

**S1 File. Supplementary materials**. Figure S1. Hybrid Bayesian network without blacklist and excluding some demographics. Figure S2. Hybrid Bayesian network with blacklist and including all demographics. Figure S3. Hybrid Bayesian network without blacklist and including all demographics. Figure S4. Averaged hybrid Bayesian network with blacklist excluding some demographics. Figure S5. Path analysis of Hybrid Bayesian network without blacklist with excluding some demographics. Figure S6. Path analysis of hybrid Bayesian network with blacklist with all demographics. Figure S7. Path analysis of hybrid Bayesian network without

blacklist with all demographics. Figure S8. Centrality statistics Bayesian network with blacklist excluding some demographics. Table S1. Centrality statistics variable codes.
(DOCX)

**S1 Data. Data_Ready_for_Analysis.**
(XLSX)

**S2 Data. ASD_2023_Text.**
(XLSX)

**S3 Data. ASD_2023_Numbers.**
(XLSX)

## Author contributions

**Conceptualization:** Daniel Shepherd, Richard J. Siegert, Jason Landon.

**Data curation:** Khan Buchwald, Matthieu Vignes.

**Formal analysis:** Khan Buchwald, Matthieu Vignes.

**Investigation:** Daniel Shepherd, Richard J. Siegert.

**Methodology:** Daniel Shepherd, Richard J. Siegert.

**Project administration:** Daniel Shepherd, Jason Landon.

**Resources:** Matthieu Vignes.

**Software:** Matthieu Vignes.

**Supervision:** Richard J. Siegert.

**Visualization:** Matthieu Vignes.

**Writing – original draft:** Daniel Shepherd, Jason Landon.

**Writing – review & editing:** Khan Buchwald, Daniel Shepherd, Richard J. Siegert, Matthieu Vignes, Jason Landon.

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
