## [Decision Letter · Decision Letter 0]

19 Nov 2024

PONE-D-24-24671Factors Predicting Parenting Stress in the Autism Spectrum Disorder Context: A Network Analysis Approach.PLOS ONE

Dear Dr. Shepherd,

Thank you for submitting your manuscript to PLOS ONE. I have now received three reviews from experts in the field. As you can see, there was some disagreement between the reviewers. After careful consideration, it seems that two of the reviewers thought there was sufficient merit to the manuscript, and I agree with this assessment. Therefore, I invite you to submit a revised version of the manuscript that addresses the points raised during the review process.

We look forward to receiving your revised manuscript.

Kind regards,

Eric J. Moody, Ph.D.

Academic Editor

PLOS ONE

Journal Requirements:

2. Peer review at PLOS ONE is not double-blinded (https://journals.plos.org/plosone/s/editorial-and-peer-review-process ). For this reason, authors should include in the revised manuscript all the information removed for blind review.

3. In the online submission form, you indicated that [The data underlying the results presented in the study are available from corresponding author (daniel.shepherd@aut.ac.nz )].

5. Please include captions for your Supporting Information files at the end of your manuscript, and update any in-text citations to match accordingly. Please see our Supporting Information guidelines for more information: http://journals.plos.org/plosone/s/supporting-information .

Reviewers' comments:

Reviewer's Responses to Questions

**Comments to the Author**

1. Is the manuscript technically sound, and do the data support the conclusions?

Reviewer #1: Yes

Reviewer #2: No

Reviewer #3: Yes

2. Has the statistical analysis been performed appropriately and rigorously? 

Reviewer #1: Yes

Reviewer #2: No

Reviewer #3: Yes

3. Have the authors made all data underlying the findings in their manuscript fully available?

Reviewer #1: Yes

Reviewer #2: No

Reviewer #3: Yes

4. Is the manuscript presented in an intelligible fashion and written in standard English?

Reviewer #1: Yes

Reviewer #2: No

Reviewer #3: Yes

5. Review Comments to the Author

Reviewer #1: Thank you very much for your invitation to review this exciting manuscript. After carefully reading this work, I indicate that this manuscript has merit and novelty. This manuscript is well-written, has a clear scope, and has high-standard scientific writing. Very good effort by the authors!

Introduction: This section is well-prepared and raises the research gap. Moreover, the authors indicate the novelty of their manuscript. The authors use appropriate references to describe the background. However, there are a few suggestions to enhance this manuscript regarding the missing essential references.

Hsiao, Y. J., Higgins, K., Pierce, T., Whitby, P. J. S., & Tandy, R. D. (2017). Parental stress, family quality of life, and family-teacher partnerships: Families of children with autism spectrum disorder. Research in developmental disabilities, 70, 152-162.

Papadopoulos A, Siafaka V, Tsapara A, et al. Measuring parental stress, illness perceptions, coping and quality of life in families of children newly diagnosed with autism spectrum disorder. BJPsych Open. 2023;9(3):e84. doi:10.1192/bjo.2023.55

Pisula, E., & Porębowicz-Dörsmann, A. (2017). Family functioning, parenting stress and quality of life in mothers and fathers of Polish children with high functioning autism or Asperger syndrome. PloS one, 12(10), e0186536.

Dijkstra-de Neijs, L., Boeke, D. B., van Berckelaer-Onnes, I. A., Swaab, H., & Ester, W. A. (2024). Parental Stress and Quality of Life in Parents of Young Children with Autism. Child Psychiatry & Human Development, 1-15.

Ilias, K., Cornish, K., Kummar, A. S., Park, M. S. A., & Golden, K. J. (2018). Parenting stress and resilience in parents of children with autism spectrum disorder (ASD) in Southeast Asia: A systematic review. Frontiers in psychology, 9, 280.

Papadopoulos, A., Fouska, S., Tafiadis, D., Trimmis, N., Plotas, P., & Siafaka, V. (2023). Psychometric Properties of the Greek Version of the Autism Parenting Stress Index (APSI) among Parents of Children with Autism Spectrum Disorder. Diagnostics, 13(20), 3259.

Methodology:

The methodology is well-presented in all the subsections. The authors provide all the required information so other researchers can replicate this study.

Results: The authors detailed this section with many tables and figures.

Discussion:

The authors discussed the results of the study appropriately and adequately. They connect their findings with other studies. However, the authors should also add the above references suggested in their discussion.

Reviewer #2: This paper, entitled “Factors Predicting Parenting Stress in the Autism Spectrum Disorder Context: A Network Analysis Approach," was submitted to PLOS ONE. It is not recommended for publication. Good luck for next time.

Reviewer #3: It is recommended to include the dataset in a public repository or as supplementary material to enhance transparency and reproducibility.

The sample's demographic bias (e.g., predominance of European mothers) limits generalizability. Discussing this limitation in more depth would strengthen the manuscript.

6. PLOS authors have the option to publish the peer review history of their article (what does this mean? ). If published, this will include your full peer review and any attached files.

**Do you want your identity to be public for this peer review?** For information about this choice, including consent withdrawal, please see our Privacy Policy .

Reviewer #1: No

Reviewer #2: No

Reviewer #3: No

---

## [Decision Letter · Decision Letter 1]

27 Jan 2025

Factors Predicting Parenting Stress in the Autism Spectrum Disorder Context: A Network Analysis Approach.

PONE-D-24-24671R1

Dear Dr. Daniel Shepherd,

We’re pleased to inform you that your manuscript has been judged scientifically suitable for publication and will be formally accepted for publication once it meets all outstanding technical requirements.

Kind regards,

Asem Surindro Singh, Ph.D

Academic Editor

PLOS ONE

Additional Editor Comments (optional):

Reviewers' comments:

Reviewer's Responses to Questions

**Comments to the Author**

1. If the authors have adequately addressed your comments raised in a previous round of review and you feel that this manuscript is now acceptable for publication, you may indicate that here to bypass the “Comments to the Author” section, enter your conflict of interest statement in the “Confidential to Editor” section, and submit your "Accept" recommendation.

Reviewer #1: All comments have been addressed

Reviewer #2: All comments have been addressed

Reviewer #3: All comments have been addressed

Reviewer #4: All comments have been addressed

2. Is the manuscript technically sound, and do the data support the conclusions?

Reviewer #1: Yes

Reviewer #2: No

Reviewer #3: Yes

Reviewer #4: Yes

3. Has the statistical analysis been performed appropriately and rigorously? 

Reviewer #1: Yes

Reviewer #2: No

Reviewer #3: Yes

Reviewer #4: Yes

4. Have the authors made all data underlying the findings in their manuscript fully available?

Reviewer #1: Yes

Reviewer #2: No

Reviewer #3: Yes

Reviewer #4: Yes

5. Is the manuscript presented in an intelligible fashion and written in standard English?

Reviewer #1: Yes

Reviewer #2: No

Reviewer #3: Yes

Reviewer #4: Yes

6. Review Comments to the Author

Reviewer #1: The authors address all the comments. Good effort by the authors. This manuscript has potential to be useful for the field.

Reviewer #2: This paper, entitled “Factors Predicting Parenting Stress in the Autism Spectrum Disorder Context: A Network Analysis Approach," was submitted to PLOS ONE. It is not recommended for publication. Good luck for next time.

Reviewer #3: (No Response)

Reviewer #4: Very nice article to read with important observations on parental stress to raise an autistic child in a family. Authors have satisfactorily addressed all the queries and taken appropriate measures in the revised article. The outcome of the article has potential social impact that can improve the quality of life of families having autistic children if presented in a more simplistic way to the affected families through public outreach activities or awareness programs.

7. PLOS authors have the option to publish the peer review history of their article (what does this mean? ). If published, this will include your full peer review and any attached files.

**Do you want your identity to be public for this peer review?** For information about this choice, including consent withdrawal, please see our Privacy Policy .

Reviewer #1: No

Reviewer #2: No

Reviewer #3: **Yes: ** Prof Dr Saad Alatrany

Reviewer #4: No

---

## [Editor Report · Acceptance letter]

PONE-D-24-24671R1

PLOS ONE

Dear Dr. Shepherd,

I'm pleased to inform you that your manuscript has been deemed suitable for publication in PLOS ONE. Congratulations! Your manuscript is now being handed over to our production team.

Kind regards,

on behalf of

Dr. Asem Surindro Singh

Academic Editor

PLOS ONE